# Investment modeling for scalable agricultural learning

Norman Peter Reeves[1], Rebecca Pietrelli[2], Ian Brooks[3], Victor G. Sal y Rosas Celi[4], Kumpati Narendra[5], Jean C. Ngabitsinze[6,7], Maximo Torero Cullen[2], Anne N. Lutomia[8], John W. Medendorp[9], Julia M. Bello-Bravo[8], Barry R. Pittendrigh[9]*

1 Sumaq Life LLC, Lansing, Michigan, United States of America, 2 Food and Agriculture Organization of the United Nations, Rome, Italy, 3 Center for Health Informatics, University of Illinois, Illinois, United States of America, 4 Sección de Matemáticas, Departamento de Ciencias, Pontificia Universidad Católica del Perú, Lima, Perú, 5 Department of Electrical Engineering, Yale University, New Haven, Connecticut, United States of America, 6 University of Rwanda, College of Agriculture, Animal Sciences and Veterinary Medicine, Rwanda, 7 Ministry of Trade and Industry, Government of Rwanda, Kigali, Rwanda, 8 Department of Agricultural Sciences Education and Communication, Purdue University, West Lafayette, Indiana, United States of America, 9 The Urban Center, Department of Entomology, Purdue University, West Lafayette, Indiana, United States of America

* barrypittendrigh@gmail.com

## Abstract

With the rise of information and communication technologies, localized farmer training can be transformed into scalable strategies applicable across diverse communities, cultures, and languages. However, the economic value of these approaches and the factors shaping their returns remain underexplored. This study presents a general framework for evaluating the economic impact of scalable agricultural learning initiatives, using multilingual instructional animations and YouTube dissemination as a case study. Systems modeling was used to simulate potential returns, assess key drivers of impact, and estimate the number of farmers required for economic viability. Sensitivity analysis shows that returns are most influenced by the cost to inform an individual, adoption rates, and income gains, and to a lesser degree, technique-sharing rates and adoption costs. When existing educational content is adapted and its lifespan extended, learning initiatives can be economically viable with few targeted farmers, making the linguistic adaption into minority or rarer languages an economically viable option. The wide variation in returns across scenarios highlights the importance of tailoring models to specific contexts to obtain more precise estimates of economic impact. These findings underscore the value of adaptable and durable learning materials and suggest that future research-for-development (R4D) investments could benefit from systems modeling to identify and prioritize high-impact agricultural solutions.

An investment into knowledge pays the best interest.

~ Benjamin Franklin

**Data availability statement:** These are simulation modeling. There is no specific raw data.

**Funding:** This work was funded in part by funds provided to JBB and BRP by Purdue University. This work was funded (BRP and JWM) in part by the United States Agency for International Development (USAID) under Agreement No. 7200AA18LE00003 as part of Feed the Future Innovation Lab for Legume Systems Research. This work was funded in part as part of the SAWBO RAPID research, supported by the United States Agency for International Development (USAID) under the terms of the agreement (BRP, JWM, and JBB). Any opinions, findings, conclusions, or recommendations expressed here are those of the authors alone and do not necessarily reflect the views or policies of USAID or the Food and Agriculture Organization of the United Nations. The funders had no role in study design, data collection and analysis, decision to publish, or preparation of the manuscript.

**Competing interests:** The authors have declared that no competing interests exist.

## Introduction

Improving agricultural practices yields significant societal benefits in developing countries, fostering both individual financial gains and national advancement [1,2]. However, challenges persist in disseminating information, notably due to literacy and language barriers prevalent in these regions [3]. To address these barriers, the utilization of Information and Communication Technologies (ICTs) in the form of video-based content and multilingual instructional animations has emerged as a promising solution [4,5]. These ICTs not only mitigate barriers but can also leverage social media platforms such as YouTube, Facebook, and WhatsApp or specialized smartphone applications for widespread dissemination [6–8]. Enhancing mobile phone usage, particularly with smartphones, has the potential to provide not just market information but also more advanced agricultural extension services [9–11]. Early research suggests that in-person training supplemented by ICTs leads to learning gains and the adoption of taught material [4,12–16] as well as sharing of knowledge [13,17].

Implementing large-scale agricultural learning initiatives often entails a significant investment [18], one that may be challenging to justify in the absence of evidence demonstrating positive returns [19]. Furthermore, the returns on this investment can vary widely depending on the nature of the solution being advocated [1]. While some initiatives may generate substantial and immediate returns, others may prove economically unfeasible. Given the complexity in measuring impacts, identifying which solutions to promote based on economic viability is not straightforward. Hence, there is a critical need for quantitative approaches to evaluate the economics of agricultural learning, to generate well-grounded estimates of returns to gauge economic viability.

The primary goal of this study is to develop a general framework for evaluating the economics of learning initiatives using a systems approach. Our work builds on a growing body of system-based applications in agriculture and natural resource (AGNR) management. Prior studies have emphasized the value of system dynamics for addressing complex AGNR problems, particularly in identifying trade-offs between short- and long-term strategies [20]. Other work has demonstrated the utility of systems modeling for evaluating technology adoption and diffusion processes in agribusiness [21], assessed policy instruments such as payment for environmental services in livestock systems [22], and developed integrated agroecosystem models that capture environmental, economic, and social trade-offs [23]. More relevant, the need for next-generation agricultural system models that leverage advances in ICT to improve decision-making and impact assessment has also been suggested [24]. Aligning with this, the present study is intended as a preliminary exploration to generate early insights into the key drivers that enhance returns in Research-for-Development (R4D) on ICT-based learning initiatives. It represents a foundational step in an iterative process that includes the design of future data collection efforts to improve model accuracy and the use of refined models to identify and prioritize high-impact agricultural learning strategies.

This paper makes three novel contributions to the literature on digital agriculture and development economics. First, we introduce a systems-modeling framework that

explicitly integrates information diffusion dynamics with paid advertisement inputs, enabling the joint analysis of organic knowledge sharing and targeted promotion in ICT-based agricultural learning programs. Second, the framework provides a quantitative evaluation of scalability and multilingual deployment strategies, allowing decision-makers to assess how language diversification with content replication influence reach, adoption, and cost-effectiveness at scale. Third, we deliver a policy-relevant economic assessment that generates comparable return-on-investment metrics, facilitating direct comparison between ICT-enabled learning initiatives and traditional agricultural extension models. Together, these contributions extend prior work by moving beyond descriptive evaluations and offering an integrated, quantitative approach for designing, scaling, and economically justifying digital agricultural learning interventions.

## Materials and methods

### Overall system

In Fig 1, we present the basic system diagram that will be used to evaluate the economic impact of learning initiatives. This systems methodology is flexible and can be adapted for a variety of interventions. For example, the system output could be economic gains from improved grain storage that minimize post-harvest loss or another key metric, such as caloric intake in a nutritional program. Additionally, this methodology can be applied to various information dissemination approaches (e.g., paper-based materials, farmer field schools). Here, we focus on multilingual instructional animations using YouTube dissemination. Given the limited understanding of the economic benefits of this dissemination approach, this type of investigation is needed.

The overall system being modelled for the agricultural learning initiative represents a series of processes that convert various system inputs into outputs. The first process, *production*, converts educational concepts and production costs into scientifically vetted learning material, such as multilingual animations that provide simple step-by-step instructions on agricultural best practices [4]. In the second process, *deployment*, animations then serve as inputs along with deployment costs (e.g., cost to promote learning concepts through YouTube) to disseminate the material to convert non-informed end-users into informed end-users. In the third process, *adoption*, informed end-users along with adoption costs serve as inputs to estimate the output, in this case, the expected revenue increase for the informed population. Finally, all the input costs for production, deployment, and adoption are fed into the last process, *returns*, along with the expected revenue increase, to estimate the internal rate of return (IRR) for the learning initiative – the overall system output.

### Production base model

In this case study, multilingual instructional animations are used to illustrate the learning concepts. To accommodate regional linguistic diversity, voice overs are used to produce language-specific versions of animations, in this case 20 language variants. An initial cost for animation creation of $20,000, with an additional expenditure of $500 per language

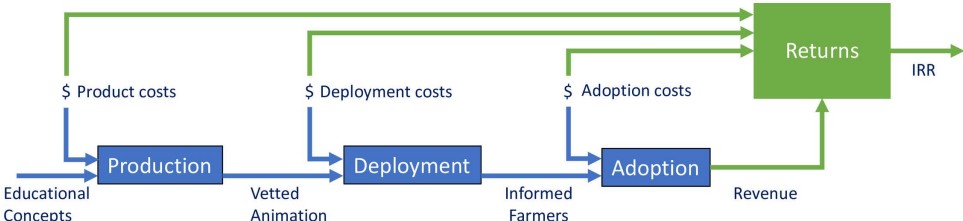

**Fig 1. Basic systems diagram illustrating the economic impact assessment of scalable agricultural learning initiatives.** The blue boxes represent generic processes that convert subsystem inputs (e.g., costs) into outputs. Note that some outputs serve as inputs for subsequent subsystems. The green box represents the process to estimate the internal rate of return (IRR), which also serves as the overall system output.

variant, is used for the base model. These are typical costs to create SAWBO multilingual instructional animations, specifically developed to inform low-literacy and non-English speaking populations [25,26].

## Deployment base model

Deployment is one of the more complex subsystems to represent, with numerous potential modeling approaches to map out information dissemination. For a comprehensive review of various model types, one can refer to [27]. For this case study, we modified a simple epidemiological model. These models, traditionally employed in the study of infectious disease, offer insights relevant to the spread of information within a population. Just as contagious diseases propagate through communities, information can exhibit similar behavior, a phenomenon effectively captured by an SI (Susceptible-Infectious) model. In our case, the SI (Susceptible-Informed) model captures the sigmoid trajectory in information dissemination, where there is some initial exponential growth, followed by a leveling off of the number of informed people over time as learning in the population becomes saturated. Fig 2 visually depicts this property of information dissemination.

Based on the simple SI model, the dynamic difference equations for information dissemination are presented in eq 1–2:

$$S(t) - S(t-1) = -aS(t-1)I(t-1) \tag{1}$$

$$I(t) - I(t-1) = aS(t-1)I(t-1) \tag{2}$$

where $S(t)$ represents the number of susceptible people, people who would want to acquire the information at year $t$, with $S(0)$ representing the initial size of the target population for information dissemination, $I(t)$ represents the number of informed people, people who have acquired the information at year $t$, with $I(0)$ representing the initial informed population, and $a$ represents a sharing effect, more specifically, the probability that an interaction between $S$ and $I$ will result in a new informed person. $(t-1)$ represents data from the previous year.

The rate of change in the number of new informed, $I(t) - I(t-1)$, is not constant but changes over time, thus giving the sigmodal shape in Fig 2. This change in the number of informed is governed by the sharing of knowledge within the system, $aS(t-1)I(t-1)$, and represents growth that occurs naturally within the population without external input (i.e., learning investments to increase the number of informed).

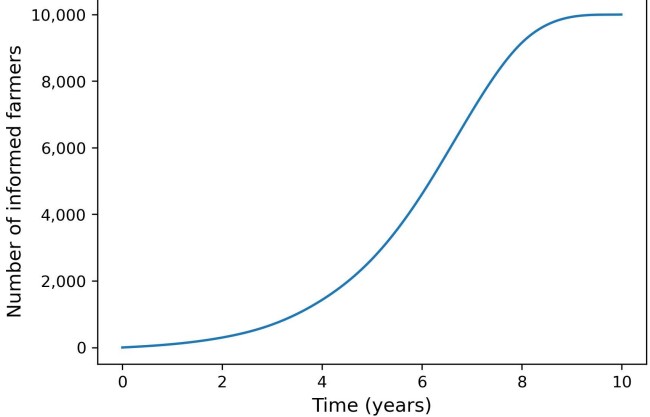

**Fig 2. Simulation of information dissemination.** The Susceptibility-Informed (SI) model captures the change in the number of informed famers over time as a result of sharing of knowledge. In this example, 10% of a population of 100,000 farmers learn the presented material.

Obtaining an initial estimate of *a* for the SI model is not straightforward (see supplemental material for details). We estimated *a*, the probability that an interaction between a susceptible and an informed individual results in a new informed individual, to be approximately 0.0001, equivalent to one new informed person per 10,000 such interactions. For ICT-based dissemination, each informed person was estimated to generate about four shares per year [7]. Accordingly, *a* increases to 0.0004 (4 × 0.0001).

Next, we expand the SI model to also capture growth in the number of informed attributed to information campaign spending, $bM(t-1)$. The expanded dynamic difference equations incorporating this growth are presented in eq 3–4:

$$S(t) - S(t-1) = -aS(t-1)I(t-1) - bM(t-1) \tag{3}$$

$$I(t) - I(t-1) = aS(t-1)I(t-1) + bM(t-1) \tag{4}$$

where $M(t-1)$ represents the amount of money in USD spent on a learning initiative in the previous year, *b* reflects the increase in the number of informed per USD spent. For online learning using multilingual instructional animations and YouTube dissemination, parameter *b* averaged 0.56 new informed per USD, which represents a cost of approximately $1.80 to obtain one new informed person [8] (Table 1). For simplicity, we assume that those exposed to the learning content have acquired the knowledge based on the high recall rate, which is typically greater than 90% for this dissemination approach based on previously published results [13].

## Adoption base model

Not all informed people will adopt the technique in the learning initiative; therefore, another parameter was included reflecting the adoption rate. For the base model, an *adoption rate* of 0.055 is used. This rate is derived from the reported average adoption rate of 49.8% for in-person training [32], adjusted to account for an estimated 11% knowledge uptake and adoption when transitioning from in-person training to mass dissemination methods (49.8% × 11% ≈ 5.5% or 0.055) [33] (Table 1).

In addition to *adoption rate*, two additional parameters are included to estimate the expected increase in income, *revenue*(*t*), in the informed population at time *t*: *annual income*, the annual income that a non-informed farmer is expected to generate, and *income gain*, the change in income from adopting the technique in the learning initiative, expressed as a proportion of the non-informed income. For the base model, the average annual income is set at $1,177 based on [34]. The income gain is assumed to be 0.19 (19%) based on metanalysis in [1] (Table 1). The revenue equation, incorporating adoption rate, annual income and income gain, is presented in eq 5:

$$revenue(t) = I(t) * adoption\ rate * annual\ income * income\ gain \tag{5}$$

**Table 1. Key model parameters based on values in agricultural learning literature.**

| Model Parameter | Base Model Value | Literature |
|---|---|---|
| Sharing rate (a) | 0.0004 | Meta analysis shows weak diffusion [1]; systemic review shows positive diffusion effect [28]; RCT shows intergenerational knowledge transfer [17]; ~ 3 non-trained farmers per trained farmers [29] |
| Number of informed per USD (b) | 0.56 ≈ $1.80 per new informed | $1.80 per new informed using SAWBO animation and YouTube [8] (range = $0.10 − $5.20, unpublished); $9 to $35 per farmer for farmers field school [30,31] |
| Adoption rate | 49.8% x 11% ≈ 5.5% | 49.8% adoption rate for SAWBO in-person training [32]; 11% knowledge uptake and adoption from in-person training to mass media campaigns [33] |
| Income gain | 19% | Meta analysis shows average increase in profits of 19% [1] (range = 11%−110%, [1,19,34–36]) |

## Returns base model

To estimate the expected IRR returns, cashflows were obtained during base model simulations over a 3-year period. This timeframe was chosen to ensure models do not vary significantly over the simulation period and to align with potential funders' preference for demonstrating immediate impact. The equation for *cashflow*(*t*) [eq 8] is derived as follows:

$$cash\ in(t)\ =\ revenue(t) \tag{6}$$

$$cash\ out(t) =\ production\ cost(0) + M(t) +\ adoption\ costs(t) \tag{7}$$

$$cashflow(t)\ =\ cash\ in(t)\ -\ cash\ out(t) \tag{8}$$

where *production costs*(0) represents the cost to create the educational animation along with the 20 language variants at year 0, $30,000 USD for the base model; *M*(*t*) represents the learning initiative campaign spending, which for the base model was set at $10,000 USD for the first 3 years; *adoption costs*(*t*) represents the yearly cost for every informed farmer who adopts the technique in the learning initiative. In this model, the cost per farmer who adopts the taught technique is anticipated to be $12 USD annually [37]. The focus here is on costs at the lower range that may be more feasible for a smallholder farmer to adopt (e.g., PICS bags), as opposed to larger capital investments that may be more suitable for larger farms.

Using cashflows, IRRs were obtained using the npf.irr(cashflow) function from the numpy_financial library in Python 3.9. Note that all cashflows represent present day values.

## Sensitivity analysis using a population of 100,000

The base model assumes fixed parameters, yet farming contexts and populations are inherently diverse. In reality, factors such as literacy levels, cultural norms, prior exposure to digital tools, socioeconomic conditions, crop types, and baseline soil quality can all lead to substantial variation in outcomes. To account for this heterogeneity, we conducted simulations across a wide range of parameter values to identify the contextual conditions under which positive returns are most likely to occur. Following standard deterministic sensitivity analysis practice using relative (multiplicative) parameter variation [38], we varied key system inputs (i.e., production cost, deployment cost, adoption cost) and parameters (i.e., sharing rate, adoption rate, annual income, income gain) by 1/4×, 1/2×, 2×, and 4× of their base model values. These ranges were selected to assess the relative sensitivity of IRR to fluctuations in the model rather than to define absolute uncertainty bounds. Because empirical variances and defensible distributions are not well characterized, relative scaling provides a transparent approach for comparing input and parameter influence without relying on speculative assumptions. In addition to IRR, the expected overall cost and revenue over the 3-year period, time to inform 10% of the population, and number of adopters were also estimated.

## Sensitivity analysis varying population size

To evaluate the economic feasibility of a learning initiative, a breakeven analysis was performed. The base model was used to estimate the minimum number of targeted farmers required to achieve a 0% IRR for target populations, *S(0)*, ranging from 1 to 10,000,000 in increments of 100. Additionally, the maximum IRR across all target population sizes was estimated.

Simulations were conducted to evaluate the effects of reducing production, deployment, and adoption costs on breakeven points and maximum IRR. Input costs (i.e., production cost, deployment cost, adoption cost) were varied by 1/4×, 1/2×, 2×, and 4× of their base model values to evaluate their impact on breakeven and maximum IRR estimates

for each scenario. For deployment, cost reductions reflect more efficient campaigns (i.e., a change in parameter *b*), rather than a reduction in campaign spending (i.e., a change in variable *M*).

### Sensitivity analysis varying the impact period

Simulations were conducted to estimate breakeven points and returns for the base model across different impact periods, ranging from 3 (base model), 5, 10, 15, and 30 years. This analysis examines how extending the timeframe affects both the maximum return and the minimum number of farmers that must be targeted for the learning initiative to remain economically viable.

All simulations were conducted in the Jupyter Notebook environment (v7.3.2) [39].

## Results

### Sensitivity analysis using a population of 100,000

Simulations using the base model predict it will take approximately 1.6 years to inform 10% of the population and to have 548 farmers adopt the taught technique for a target population of 100,000 (Fig 3). The expected total cost over the 3-year period is projected to be $84,098, which is inclusive of the $30,000 investment in the instructional animation creation with 20 languages ($20,000 for the original animation + $10,000 in translations), $30,000 in campaign spending, and $12 per year costs for the farmer to employ the technique. The projected revenue generated from an increase in income over the 3-year period is approximately $449,217. The returns under a base model scenario for the agriculture learning initiative are expected to generate an IRR of 208%.

Sensitivity analysis revealed that returns were more influenced by production and deployment costs than adoption costs (Fig 4). Interestingly, reducing annual campaign spending (M), led to lower returns, whereas more aggressive campaign spending yielded higher economic benefits. Among the key parameters, returns were most sensitive to cost per informed, adoption rates, annual income, and income gain, while adoption costs and sharing rates had a comparatively smaller impact.

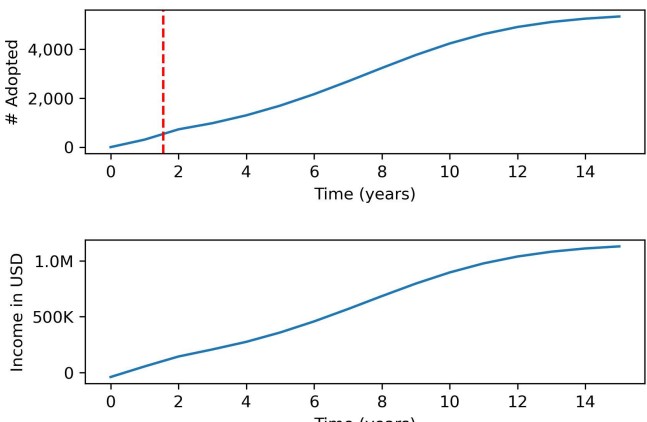

**Fig 3. Simulation using the base model.** The top panel shows the projected increase over time in the number of farmers expected to adopt the taught technique. The dotted vertical line represents the period when 10% of population is informed. The bottom panel illustrates the corresponding increase in revenue resulting from farmers' adoption of the technique.

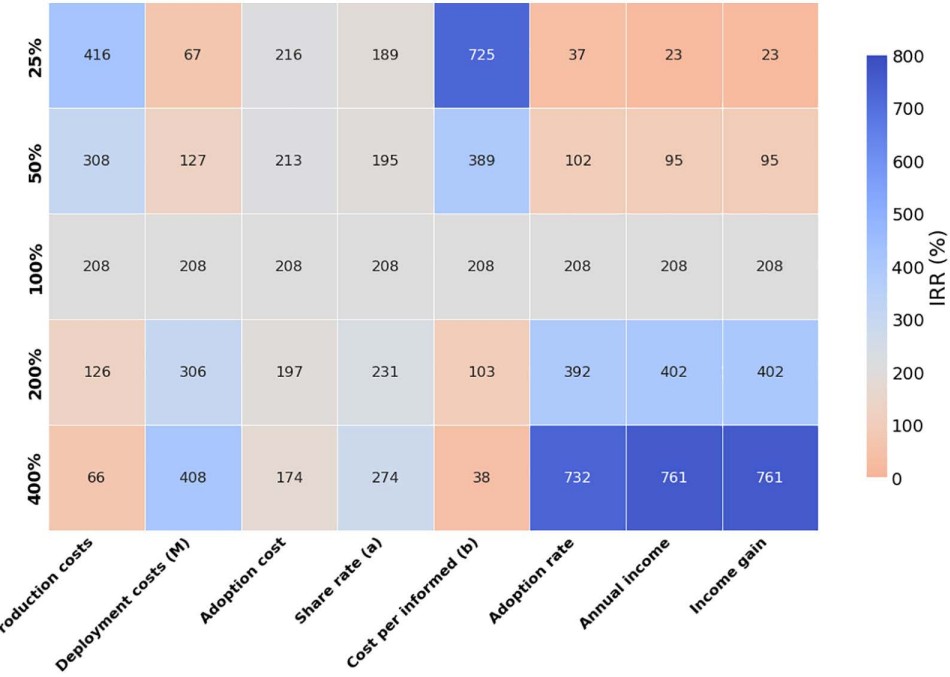

**Fig 4. Heatmap visualizing the influence of base model changes on returns.** Relative changes of 1/4×=25%, 1/2×=50%, 1×=100% (base model), 2×=200%, and 4×=400% were simulated. Higher returns are represented in blue and lower returns in red, with a neutral reference point set at the base model return of 208%.

## Sensitivity analysis varying population size

As the number of targeted farmers increased, the expected returns also increased monotonically, reaching a peak of approximately 651% in the base model scenario (Fig 5; Table 2). Adjusting production costs had little impact on the maximum return, which remained within a narrow range of 606% to 664% (Fig 5 left, see 3-year impact period in Table 2). In contrast, changes in the cost per informed had a substantial effect, with maximum returns ranging from 152% at $7.20 per new informed to 2,538% at $0.45 per new informed (Fig 5 center). Adoption costs, however, had a minimal influence on the maximum return, which varied only slightly between 542% and 678% (Fig 5 right).

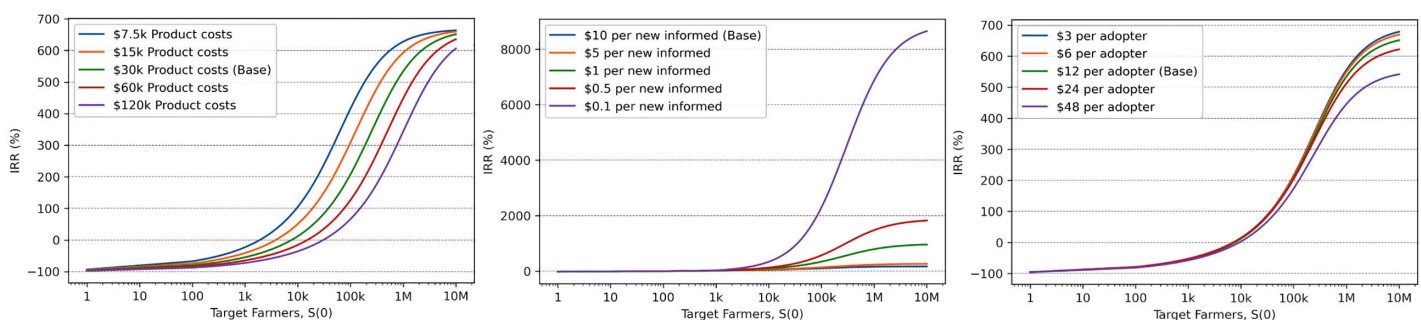

**Fig 5. The effect of target population size on internal rate of returns (IRR).** Increasing the target population size (S(0)), which represents the number of farmers who would benefit from the taught technique, led to higher returns. The left plot shows how reducing production costs affects returns slightly. The center plot reveals that lower deployment costs (i.e., more cost-effective campaigns) can substantially enhance returns for larger target populations. The right plot indicates that reductions in adoption costs have a minimal impact on returns.

**Table 2. Influence of production costs and impact period on the number of farmers needing to be targeted to breakeven and the max IRR using a target population of 10 million farmers.**

| Production Costs | Number of farmers to be targeted to breakeven | | | | | Max IRR | |
|---|---|---|---|---|---|---|---|
| | 3 years | 5 years | 10 years | 15 years | 30 years | 3 years | 30 years |
| $120k (400%) | 30,370 | 11,664 | 2,707 | 1,228 | 445 | 606% | 612% |
| $60k (200%) | 15,185 | 5,832 | 1,354 | 615 | 224 | 635% | 641% |
| $30k (100%) | 7,593[a] | 2,916 | 678 | 308 | 113 | 651% | 657% |
| $15k (50%) | 3,796 | 1,459 | 341 | 160 | 83 | 659% | 665% |
| $7.5k (25%) | 1,898 | 730 | 174 | 93 | 64 | 664% | 669% |
| $500 (1.7%)[b] | 130 | 74 | 39 | 24 | 7 | 667% | 673% |

[a]Base model scenario of $30k production cost with impact period of 3 years.

[b]A typical cost to develop a new language variant for multilingual instructional animations is approximately $500.

When evaluating the breakeven point for the learning investment, the number of farmers that needed to be targeted to generate a positive return was more sensitive to production and deployment costs than adoption costs. Reducing production costs from the base model value of $30,000 to $7,500 lowered the breakeven point from 7,593 farmers to 1,898 farmers (Table 2). Similarly, decreasing deployment costs from $1.80 (b = 0.56) to $0.45 (b = 2.2) per new informed reduced the breakeven point from 7,593 farmers to 1,950 farmers. In contrast, reducing the adoption costs only marginally shifted the breakeven point to 7,260 farmers under the best-case scenario.

## Sensitivity analysis varying the impact period

All impact periods showed similar maximum returns but varied in terms of breakeven points (Fig 6; Table 2). For economic viability, shorter timeframes require targeting substantially larger farmer populations: 77,592 farmers for a 3-year period and 22,916 farmers for a 5-year period. This contrasts with longer impact periods, where viable target populations are around 678 farmers for a 10-year period, 308 farmers for a 15-year period, and 113 farmers for a 30-year period.

## Discussion

The primary goal of this study was to develop a general framework for evaluating the economic impact of scalable agricultural learning initiatives using a systems approach. Specifically, we employed system models to simulate potential returns on ICT-based learning investments, analyze factors influencing returns, and predict the number of farmers needing to be targeted to breakeven to achieve economic viability. Preliminary simulations suggest that ICT-based learning initiatives aimed at improving agricultural practices can yield substantial returns and appear for most scenarios to be economically viable. However, the considerable range in estimates across various scenarios underscores the necessity for tailored modelling of specific learning initiatives to verify their economic benefits.

Simulations highlight several key factors for maximizing returns. Selecting learning topics that lead to high adoption rates, drive income gains, and lower the cost per informed farmer, is essential, as these are critical determinants of overall returns. Further, increasing early campaign spending accelerates adoption, generating the revenue needed to offset initial production costs more efficiently. While scaling outreach to larger audiences significantly enhances returns, targeting smaller farmer groups can also be viable if production costs are minimized. In this regard, multilingual instructional animations offer a cost-effective approach, enabling content to be shared across multiple regions. Follow-up simulations indicate that producing an additional language variant for $500 would require only 130 farmers to breakeven (Fig 7 and Table 2). Moreover, extending the lifespan of learning initiatives significantly reduces the number of farmers needed to achieve economic viability, making programs that address ongoing or recurring challenges, where the same content can be reused, more economically viable than those focused on transient, short-term issues.

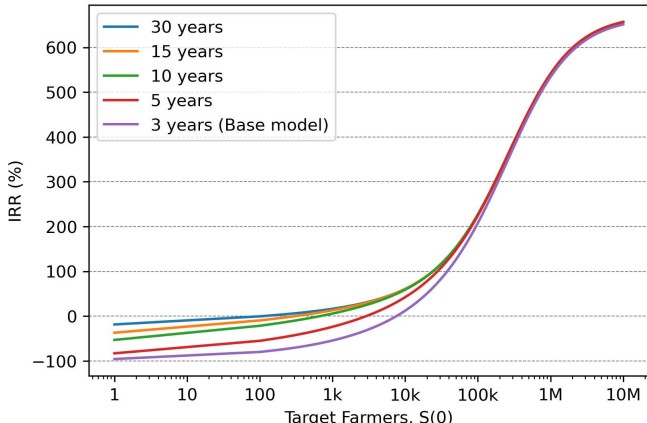

Fig 6. **Impact period influence on internal rate of return (IRR) and breakeven point.** Breakeven points represent the number of farmers needing to be targeted to cross the 0% IRR threshold. All impact periods show similar maximum returns (IRR) but varied in terms of breakeven points. Longer impact periods (10, 15, 30 years) significantly shift breakeven points to smaller target populations, indicating that short-term projects (3 and 5 years) require targeting larger farmer groups to remain economically viable.

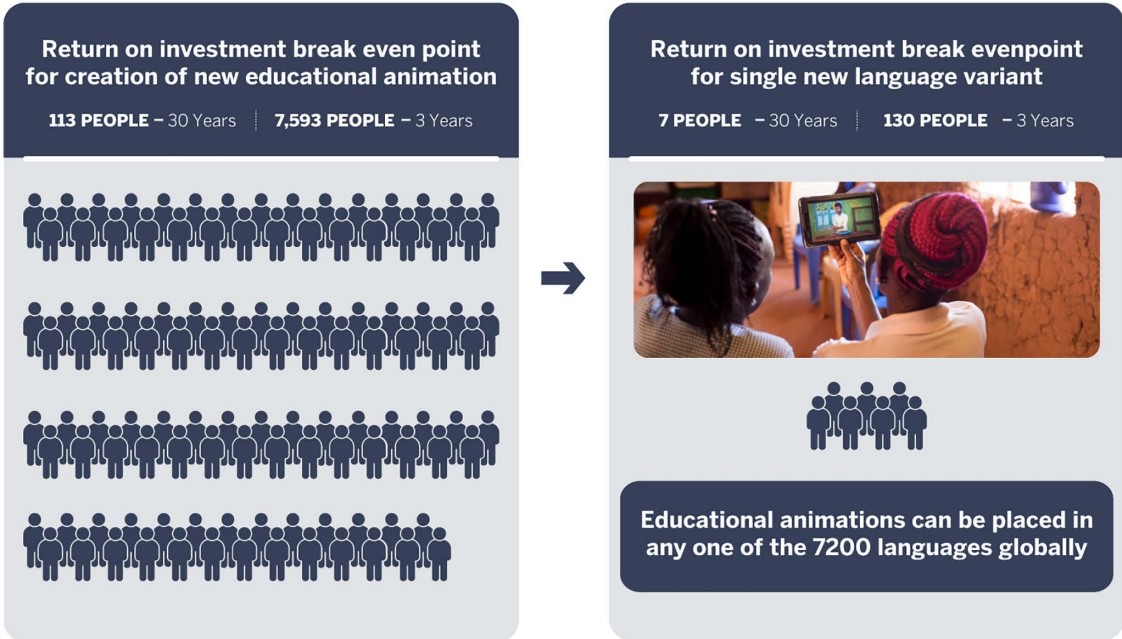

Fig 7. **Number of farmers required to breakeven.** The number of farmers required to breakeven under different scenarios for both new animation creation (left) and a new language variant (right) creation of an existing animation at 30 years or 3 years.

Our study examined a broad range of scenarios, with the base model projecting an internal rate of return (IRR) of 208% for a target population of 100,000. Notably, our 30 year IRR estimate of 657% far exceeds the reported 30 year average of 45% for agricultural learning initiatives in Africa [40]. While this discrepancy raises questions about model accuracy, it may also reflect fundamental differences in dissemination approaches. Published returns are largely derived from farmer field schools, which involve deployment costs ranging from $9 to $35 per farmer [30,31]. In contrast, the digital learning

approach modeled here assumes a substantially lower cost of $1.80 per informed individual [8] (Table 1), which may partially explain the higher projected returns. A follow-up simulation using a $35 per new informed farmer resulted in a 30-year return of 52%, aligning more closely with published estimates. Nevertheless, given the preliminary nature of our analysis, we cannot state with certainty that digital dissemination yields returns as high as those simulated, nor can we definitively identify the key drivers of these elevated returns if they do exist.

Recent research in information systems suggests that scalable digital platforms, such as large language models applied in blockchain-based supply chain finance and decentralized finance [41,42], create economic value by reducing information asymmetries, and exploiting high fixed development costs with low deployment costs [43]. Although these studies focus on financial and enterprise systems, the same economic mechanisms may underpin scalable digital agricultural learning platforms. This study extends this literature by quantitatively modeling how multilingual, digitally delivered agricultural knowledge translates these efficiencies into income gains for smallholder farmers, showing the potential for substantial economic returns.

In terms of limitations, our study permitted system parameters to vary independently in simulations, whereas in reality, these attributes are likely correlated. For instance, adoption costs and adoption rates likely exhibit interdependence. By modelling more specific scenarios, it should be possible to account for these dependencies and to generate more precise estimates. Subsequent research efforts should prioritize real-world data collection to refine models, while leveraging current model insights. Our simulations underscore the time-intensive nature of educating the population (see Fig. 2), particularly at lower campaign spending levels where sharing of learning content predominantly serves to increase awareness. Consequently, greater campaign spending is recommended not only to improve returns but also to facilitate data collection efforts for model refinement, given that a larger population of informed farmers enables more robust sampling essential for accurately measuring system characteristics. Insights gleaned from simulations can inform the design of data collection methodologies, including determining the optimal timing for sampling the population following the start of a learning intervention and identifying key variables to be precisely measured based on sensitivity analysis.

Furthermore, future research should explore alternative model types, such as social network graphs and time-varying models, to enhance predictive capabilities while upholding reliability and interpretability. As more granular data becomes available, models can evolve in complexity as needed to better mirror real-world intricacies. This could include incorporating parallel dissemination channels to reach both technology adept farmers using ICT-based systems and other farmers using more traditional extension services.

## Conclusions

Drawing lessons from past industrial revolutions, which transformed production from artisanal to scalable processes, underscores the potential for a similar transformation in agricultural learning within international development. The rise of ICTs and online dissemination platforms presents a timely opportunity to move beyond traditional training methods toward more advanced, scalable approaches. This evolution not only broadens access to learning but also reduces the cost of translating R4D innovations into practical, income-generating applications for farmers.

At the same time, several limitations of the present study warrant consideration. The simulation-based results rely on simplifying assumptions to represent complex real-world processes (e.g., information sharing, adoption behavior, and income gains) and have not been validated over long time horizons or across diverse agroecological conditions. In addition, the current analysis focuses on economic outcomes and does not explicitly account for broader environmental or societal impacts. Importantly, however, the modeling framework is designed to support iterative refinement: as ICT-based learning initiatives are deployed, empirical data on reach, adoption, and outcomes can be continuously collected to update model parameters in a context-specific manner. Viewed in this light, the simulations provide an initial decision-support baseline that can be progressively improved over time, rather than a fixed or static forecast.

 

Looking ahead, future R4D learning initiatives could benefit from the integration of systems modeling to identify, compare, and prioritize high-impact agricultural solutions. For organizations focused on building farmer capacity in international development settings, there is a critical need for decision-support tools that assess the potential value of training and learning investments. Prior to launching a learning initiative, preliminary system estimates (e.g., expected number of farmers targeted, projected increases in income, etc...) can be used to forecast economic returns. This approach helps ensure that limited resources are directed toward initiatives with the greatest potential impact.

In regions like Africa, where rapid population growth and climate change threaten food security, the urgency to scale and optimize agricultural learning is especially acute. As our simulations suggest, ICT-based approaches hold significant promise, not only for combating hunger, but also for advancing economic development among farming communities. Moreover, digital media offers several unique advantages that reinforce this potential. Its flexibility enables rapid multi-lingual adaptation and dissemination across diverse linguistic and cultural contexts; its persistence allows content to be reused over time at minimal cost; and its network effects arising from peer-to-peer sharing and social platforms further amplify the reach of digital learning, an effect captured in our SI-inspired modeling framework. Collectively, these characteristics highlight the potential of ICT-based learning to complement traditional agricultural capacity-building approaches by promoting more inclusive, efficient, and sustainable forms of knowledge diffusion in the Global South.

## Declaration of generative AI and AI-assisted technologies in the writing process

During the preparation of this work, the authors used ChatGPT 4.o to improve readability and check for grammatical errors. The authors generated the original text and then evaluated sections with the help of ChatGPT. Suggestions from ChatGPT were reviewed and edited as needed by the authors. The authors take full responsibility for the final content of the publication.

## Supporting information

**S1 File. Supplemental materials.**
(DOCX)

## Author contributions

**Conceptualization:** Norman Peter Reeves, Rebecca Pietrelli, Kumpati Narendra, John William Medendorp, Julia Bello-Bravo, Barry Pittendrigh.

**Data curation:** Norman Peter Reeves.

**Formal analysis:** Norman Peter Reeves, Victor G. Sal y Rosas Celi, Kumpati Narendra.

**Funding acquisition:** John William Medendorp, Julia Bello-Bravo, Barry Pittendrigh.

**Investigation:** Norman Peter Reeves, Rebecca Pietrelli, Ian Brooks, Victor G. Sal y Rosas Celi, Barry Pittendrigh.

**Methodology:** Norman Peter Reeves, Rebecca Pietrelli, Ian Brooks, Victor G. Sal y Rosas Celi, Kumpati Narendra, Jean Ngabitsinze, Maximo Torero Cullen, John William Medendorp, Barry Pittendrigh.

**Project administration:** Norman Peter Reeves, Barry Pittendrigh.

**Resources:** Barry Pittendrigh.

**Validation:** Norman Peter Reeves, Victor G. Sal y Rosas Celi, Jean Ngabitsinze, Maximo Torero Cullen, Anne Namatsi Lutomia.

**Visualization:** Norman Peter Reeves.

**Writing – original draft:** Norman Peter Reeves.

**Writing – review & editing:** Norman Peter Reeves, Rebecca Pietrelli, Ian Brooks, Victor G. Sal y Rosas Celi, Kumpati Narendra, Jean Ngabitsinze, Maximo Torero Cullen, Anne Namatsi Lutomia, John William Medendorp, Julia Bello-Bravo, Barry Pittendrigh.

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
