## [Decision Letter · Decision Letter 0]

7 Sep 2025

Dear Dr. Pittendrigh,

Thank you for submitting your manuscript to PLOS ONE. After careful consideration, we feel that it has merit but does not fully meet PLOS ONE’s publication criteria as it currently stands. Therefore, we invite you to submit a revised version of the manuscript that addresses the points raised during the review process.

We look forward to receiving your revised manuscript.

Kind regards,

Yang (Jack) Lu, PhD

Academic Editor

PLOS ONE

Journal Requirements:

3. Please note that PLOS One has specific guidelines on code sharing for submissions in which author-generated code underpins the findings in the manuscript. In these cases, we expect all author-generated code to be made available without restrictions upon publication of the work. Please review our guidelines at https://journals.plos.org/plosone/s/materials-and-software-sharing#loc-sharing-code and ensure that your code is shared in a way that follows best practice and facilitates reproducibility and reuse.

This work was funded in part by funds provided to JBB and BRP by Purdue University. This work was funded (BRP and JWM) in part by the United States Agency for International Development (USAID) under Agreement No. 7200AA18LE00003 as part of Feed the Future Innovation Lab for Legume Systems Research. This work was funded in part as part of the SAWBO RAPID research, supported by the United States Agency for International Development (USAID) under the terms of the agreement (BRP, JWM, and JBB). Any opinions, findings, conclusions, or recommendations expressed here are those of the authors alone and do not necessarily reflect the views or policies of USAID or the Food and Agriculture Organization of the United Nations.

6. Please note that funding information should not appear in any section or other areas of your manuscript. We will only publish funding information present in the Funding Statement section of the online submission form. Please remove any funding-related text from the manuscript.

7. Please amend your authorship list in your manuscript file to include author Barry Pittendrigh, Jean Ngabitsinze.

8. Please amend the manuscript submission data (via Edit Submission) to include author Barry Robert Pittendrigh, Jean C. Ngabitsinze.

Reviewers' comments:

Reviewer's Responses to Questions

**Comments to the Author**

1. Is the manuscript technically sound, and do the data support the conclusions?

Reviewer #1: Yes

Reviewer #2: Yes

Reviewer #3: Partly

2. Has the statistical analysis been performed appropriately and rigorously?

Reviewer #1: N/A

Reviewer #2: Yes

Reviewer #3: Yes

3. Have the authors made all data underlying the findings in their manuscript fully available?

Reviewer #1: Yes

Reviewer #2: Yes

Reviewer #3: Yes

4. Is the manuscript presented in an intelligible fashion and written in standard English?

Reviewer #1: Yes

Reviewer #2: Yes

Reviewer #3: Yes

Reviewer #1: 1. Lack of Heterogeneity and Parameter Correlation

The model assumes that key parameters such as adoption rates, income gains, and campaign costs are fixed and independent across contexts. However, in practice, these factors are often highly heterogeneous and interdependent. For example, adoption rates vary widely depending on farmers’ literacy levels, cultural norms, prior exposure to digital tools, and socioeconomic conditions. Similarly, income gains differ substantially by crop type, farming system, and baseline productivity. Furthermore, multilingual adaptation does not guarantee equal comprehension — regional dialects, education levels, and digital literacy gaps often affect knowledge transfer effectiveness. Ignoring such heterogeneity and parameter correlations risks biasing ROI estimates and overstating scalability. I recommend that the authors expand the discussion to clarify how these correlations may impact overall findings.

2. Writing Quality and Clarity

The manuscript would benefit from improved English writing to enhance clarity, especially in the methodology and results sections. Certain assumptions and parameter derivations are difficult to follow, which may limit reproducibility and interpretability. Providing clearer explanations, illustrative tables, and stronger linkage between model assumptions and referenced literature would significantly improve readability. Prior PLOS ONE studies on agricultural investment modeling demonstrate the value of transparent parameter reporting and plain-language summaries when communicating findings to broader audiences.

3. Lack of Sensitivity and Robustness Analysis

The reported 30-year IRR of 657% is extraordinarily high compared to typical ex-post evaluations of agricultural extension programs, which often report returns closer to 30–50%. While the authors attribute this to cost efficiencies enabled by digital dissemination, the manuscript lacks sufficient sensitivity analyses and robustness checks to support this conclusion. For example, how would the IRR change if adoption rates were lower, income gains were overestimated, or translation and campaign costs were higher? Including formal sensitivity testing or Monte Carlo simulations would greatly improve the credibility and policy relevance of the results.

Reviewer #2: Strengths:

1. Analyze and recognize key control components for evaluating such programs.

2. Clearly point out the problems they want to solve and define the scenarios.

3. Correct and sound mathematical deductions during the model’s set-up section.

4. Comprehensive experiments with discussion and conclusion.

Major concerns

1. Methodology: Consider using a hybrid systems model instead of purely SI approach. The SI approach could be a sub-component and think about how to integrates it with cost-benefit dynamics, various groups of farmers (with different backgrounds) and time-varying decay.

2. Writing & Presentation: Consider better highlighting Subtitles like “Adoption base model” to improve readibility.

3. Experiment & Observations: Concerns raised on equations’ parameters in Table 1, some key parameters like SAWBO animation and YouTube cost per informed, only citing one paper may cause bias.

Minor comments & suggestions

Line 170, “indicate” typesetting or typo issues.

Requests for clarification

The paper ignores any social and policy-related factors and only focuses on technical modeling. Is that by deliberation?

Reviewer #3: The manuscript presents a framework for evaluating the economic impact of scalable agricultural learning initiatives.

- While the manuscript applies a systems modeling framework, it lacks comprehensive coverage of recent developments in this area. More recent applications of system approaches could be discussed to highlight what is novel in this work.

- The model used in the article relies on digital media, but lacks the nature of digital media.

**Do you want your identity to be public for this peer review?** For information about this choice, including consent withdrawal, please see our Privacy Policy

Reviewer #1: No

Reviewer #2: No

Reviewer #3: No

---

## [Author Response · Author response to Decision Letter 1]

5 Dec 2025

The response to the reviewers is including in the PDF build (at the end of the PDF build).

---

## [Decision Letter · Decision Letter 1]

29 Dec 2025

Dear Dr. Pittendrigh,

Thank you for submitting your manuscript to PLOS ONE. After careful consideration, we feel that it has merit but does not fully meet PLOS ONE’s publication criteria as it currently stands. Therefore, we invite you to submit a revised version of the manuscript that addresses the points raised during the review process.

We look forward to receiving your revised manuscript.

Kind regards,

Yang (Jack) Lu, PhD

Academic Editor

PLOS One

Journal Requirements:

Reviewers' comments:

Reviewer's Responses to Questions

**Comments to the Author**

Reviewer #4: (No Response)

Reviewer #5: (No Response)

2. Is the manuscript technically sound, and do the data support the conclusions?

Reviewer #4: (No Response)

Reviewer #5: (No Response)

3. Has the statistical analysis been performed appropriately and rigorously?

Reviewer #4: (No Response)

Reviewer #5: (No Response)

4. Have the authors made all data underlying the findings in their manuscript fully available?

Reviewer #4: (No Response)

Reviewer #5: (No Response)

5. Is the manuscript presented in an intelligible fashion and written in standard English?

Reviewer #4: (No Response)

Reviewer #5: (No Response)

Reviewer #4: 1.The original method section specifies that "Simulations varied system inputs (i.e., production cost, deployment cost, adoption cost) and parameters (i.e., sharing rate, adoption rate, annual income, income gain) by 1/4 x, 1/2 x, 2 x, and 4 x of their base model values to evaluate their impact on overall returns." However, it does not provide the basis for selecting the range of parameter variations. It is recommended to briefly supplement this explanation

2.The original text clearly states that "to accommodate regional linguistic diversity, voice-overs are used to produce language-specific versions of animations, in this case, 20 language variants. The initial cost for animation creation is $20,000, with an additional expenditure of $500 per language variant, which is used for the base model." It also confirms the economic feasibility of multi-language adaptation through simulation, but does not specify the contribution of the 20 language variants to the overall Internal Rate of Return (IRR). It is recommended to provide additional explanations.

3.The original text mentions that "the cost of digital learning (1.8 USD per informed individual) is lower than that of farmer field schools (9-35 USD per farmer)", but the comparison of economic efficiency between the two models is briefly discussed in the conclusion. It is suggested to supplement the conclusion with data from Table 1 in the original text.

Reviewer #5: The manuscript proposes a systems modeling-based framework for assessing the economic impact of scalable agricultural learning programs. The research questions are of practical significance, however the manuscript still has several areas for improvement.

1. The contribution of the manuscript should be further elaborated in the Introduction.

2. The economic impact of the manuscript should be fully analyzed in conjunction with existing literature, such as Potential of large language models in blockchain-based supply chain finance. Enterprise Information Systems, 19(11), 2541199 and Decentralized finance (DeFi): a paradigm shift in the Fintech. Enterprise Information Systems, 18(9), 2397630.

3. It is recommended to further analyze the limitations of the manuscript in Conclusions.

**Do you want your identity to be public for this peer review?** For information about this choice, including consent withdrawal, please see our Privacy Policy

Reviewer #4: No

Reviewer #5: No

---

## [Author Response · Author response to Decision Letter 2]

6 Feb 2026

PlosOne

Point-by-Point Response to Reviewers

PONE-D-25-41099R1

Title: Investment Modeling for Scalable Agricultural Learning

Dear editor and reviewers,

We sincerely appreciate your thoughtful comments. In response to your feedback, we have made revisions to the paper, which are outlined in detail in the point-by-point response below. Our comments are in italics. Parts taken from the paper are in "quotation marks", and new sections added are "underlined".

Reviewer #4

Comment 1

The original method section specifies that "Simulations varied system inputs (i.e., production cost, deployment cost, adoption cost) and parameters (i.e., sharing rate, adoption rate, annual income, income gain) by 1/4 x, 1/2 x, 2 x, and 4 x of their base model values to evaluate their impact on overall returns." However, it does not provide the basis for selecting the range of parameter variations. It is recommended to briefly supplement this explanation.

Response 1

Our objective in the sensitivity analysis was to assess the relative change in internal rate of return (IRR) in response to fluctuations in key model parameters, rather than to estimate absolute uncertainty bounds for each parameter. Accordingly, we selected multiplicative variations (1/4 x, 1/2 x, 2 x, and 4 x of their base values) that span a sufficiently wide range to reveal the directional influence and relative sensitivity of IRR to each input. For several parameters, empirical variance estimates are not well characterized across contexts, and specifying precise probabilistic ranges would require speculative assumptions. In this setting, relative scaling provides a transparent approach for comparing parameter influence.

To account for heterogeneity, we conducted simulations across a wide range of parameter values to identify the contextual conditions under which positive returns are most likely to occur. Following standard deterministic sensitivity analysis practice using relative (multiplicative) parameter variation (Briggs et al., 2006), we varied key system inputs (i.e., production cost, deployment cost, adoption cost) and parameters (i.e., sharing rate, adoption rate, annual income, income gain) by 1/4×, 1/2×, 2×, and 4× of their base model values. These ranges were selected to assess the relative sensitivity of IRR to fluctuations in the model rather than to define absolute uncertainty bounds. Because empirical variances and defensible distributions are not well characterized, relative scaling provides a transparent approach for comparing input and parameter influence without relying on speculative assumptions.

Comment 2

The original text clearly states that "to accommodate regional linguistic diversity, voice-overs are used to produce language-specific versions of animations, in this case, 20 language variants. The initial cost for animation creation is $20,000, with an additional expenditure of $500 per language variant, which is used for the base model." It also confirms the economic feasibility of multi-language adaptation through simulation, but does not specify the contribution of the 20 language variants to the overall Internal Rate of Return (IRR). It is recommended to provide additional explanations.

Response 2

We may be misunderstanding the specific analysis requested by the reviewer. A cost-only counterfactual in which language-variant costs are removed increases the base-model IRR from 208% to 265%, but this assumes the same target population of farmers. In practice, fewer language variants would likely reduce reach, which would also reduce IRR. Our findings indicate that scaling to larger target populations is essential for improving IRR by spreading fixed production costs over a broader audience.

As part of the sensitivity analysis, we completed analysis specifically focusing on production costs, including the incremental cost of adding language variants, and their impact on both max IRR and the breakeven point. As shown in Table 2, variation in production costs has minimal effect on max IRR but substantially affects the number of farmers required to reach breakeven; specifically, adding a language variant at a cost of $500 requires reaching relatively few additional farmers (ranging from approximately 130 to 7, depending on the assumed duration of impact).

In other words, the language adaptation, and the associated costs, is a condition for increasing the number of targeted farmers. Once the language customization is in place, we can scale up the outreach. The simulation, therefore, focuses on modelling the change in the number of farmers targeted, not on the number of languages, even though the two aspects are connected. Languages determine the feasibility and cost of expansion, but the simulation itself is about estimating how the target population increases once those prerequisites are met.

"In this regard, multilingual instructional animations offer a cost-effective approach, enabling content to be shared across multiple regions. Follow-up simulations indicate that producing an additional language variant for $500 would require only 130 farmers to breakeven (Fig 7 and Table 2). Moreover, extending the lifespan of learning initiatives significantly reduces the number of farmers needed to achieve economic viability, making programs that address ongoing or recurring challenges, where the same content can be reused, more economically viable than those focused on transient, short-term issues."

Table 2. Influence of production costs and impact period on the number of farmers needing to be targeted to breakeven and the max IRR using a target population of 10 million farmers.

Production

Costs Number of farmers to be targeted to breakeven Max IRR

3 years 5 years 10 years 15 years 30 years 3 years 30 years

$120k (400%) 30,370 11,664 2,707 1,228 445 606% 612%

$60k (200%) 15,185 5,832 1,354 615 224 635% 641%

$30k (100%) 7,593a 2,916 678 308 113 651% 657%

$15k (50%) 3,796 1,459 341 160 83 659% 665%

$7.5k (25%) 1,898 730 174 93 64 664% 669%

$500 (1.7%)b 130 74 39 24 7 667% 673%

a Base model scenario of $30k production cost with impact period of 3 years.

b A typical cost to develop a new language variant for multilingual instructional animations is approximately $500.

Comment 3

The original text mentions that "the cost of digital learning (1.8 USD per informed individual) is lower than that of farmer field schools (9-35 USD per farmer)", but the comparison of economic efficiency between the two models is briefly discussed in the conclusion. It is suggested to supplement the conclusion with data from Table 1 in the original text.

Response 1

In response to this comment, we revised the Discussion to explicitly reference the cost estimates reported in Table 1 (USD 1.80 per informed individual for digital learning versus USD 9–35 per farmer for field schools), thereby grounding the comparison of economic efficiency in the quantitative results presented earlier in the manuscript.

"Published returns are largely derived from farmer field schools, which involve deployment costs ranging from $9 to $35 per farmer[30, 31]. In contrast, the digital learning approach modeled here assumes a substantially lower cost of USD 1.80 per informed individual[8] (Table 1), which may partially explain the higher projected returns. A follow-up simulation using a $35 per new informed farmer resulted in a 30-year return of 52%, aligning more closely with published estimates. Nevertheless, given the preliminary nature of our analysis, we cannot state with certainty that digital dissemination yields returns as high as those simulated, nor can we definitively identify the key drivers of these elevated returns if they do exist.

Reviewer #5

Comment 1

The contribution of the manuscript should be further elaborated in the Introduction.

Response 1

We thank the reviewer for this suggestion. The Introduction has been revised to more clearly articulate the manuscript’s contributions.

"This paper makes three novel contributions to the literature on digital agriculture and development economics. First, we introduce a systems-modeling framework that explicitly integrates information diffusion dynamics with paid advertisement inputs, enabling the joint analysis of organic knowledge sharing and targeted promotion in ICT-based agricultural learning programs. Second, the framework provides a quantitative evaluation of scalability and multilingual deployment strategies, allowing decision-makers to assess how language diversification with content replication influence reach, adoption, and cost-effectiveness at scale. Third, we deliver a policy-relevant economic assessment that generates comparable return-on-investment metrics, facilitating direct comparison between ICT-enabled learning initiatives and traditional agricultural extension models. Together, these contributions extend prior work by moving beyond descriptive evaluations and offering an integrated, quantitative approach for designing, scaling, and economically justifying digital agricultural learning interventions."

Comment 2

The economic impact of the manuscript should be fully analyzed in conjunction with existing literature, such as Potential of large language models in blockchain-based supply chain finance. Enterprise Information Systems, 19(11), 2541199 and Decentralized finance (DeFi): a paradigm shift in the Fintech. Enterprise Information Systems, 18(9), 2397630.

Response 2

We appreciate this recommendation and have expanded the Discussion section to situate our findings within the broader digital-economy and information-systems literature. While our focus differs from blockchain-based supply chain finance and decentralized finance systems, we now draw conceptual parallels regarding scalability and exploiting high fixed development costs with low marginal deployment costs. These references help contextualize our results within emerging digital-infrastructure frameworks and underscore the broader relevance of scalable information delivery systems to economic development.

"Recent research in information systems suggests that scalable digital platforms, such as large language models applied in blockchain-based supply chain finance and decentralized finance[40, 41], create economic value by reducing information asymmetries, and exploiting high fixed development costs with low deployment costs[42]. Although these studies focus on financial and enterprise systems, the same economic mechanisms may underpin scalable digital agricultural learning platforms. This study extends this literature by quantitatively modeling how multilingual, digitally delivered agricultural knowledge translates these efficiencies into income gains for smallholder farmers, showing the potential for substantial economic returns."

40. Yang L, Hou Q, Lu Y, Xu LD. Potential of large language models in blockchain-based supply chain finance. Enterprise Information Systems. 2025;19(11). doi: 10.1080/17517575.2025.2541199.

41. Romero-Castro N, López-Cabarcos MÁ, Vittori-Romero V, Piñeiro-Chousa J. Decentralized Finance in Business and Economics Research: A Bibliometric Analysis. International Journal of Financial Studies. 2025;13(4). doi: 10.3390/ijfs13040211.

42. Hoekman B, Maskus K, Stephenson M, Tajoli L, Tentori D. Leveraging global digital trade opportunities for all. Rome, Italy: T20 Italy, 2021.

Comment 3

It is recommended to further analyze the limitations of the manuscript in Conclusions.

Response 3

We agree and have strengthened the Conclusions section by explicitly discussing key limitations.

"At the same time, several limitations of the present study warrant consideration. The simulation-based results rely on simplifying assumptions to represent complex real-world processes (e.g., information sharing, adoption behavior, and income gains) and have not been validated over long time horizons or across diverse agroecological conditions. In addition, the current analysis focuses on economic outcomes and does not explicitly account for broader environmental or societal impacts. Importantly, however, the modeling framework is designed to support iterative refinement: as ICT-based learning initiatives are deployed, empirical data on reach, adoption, and outcomes can be continuously collected to update model parameters in a context-specific manner. Viewed in this light, the simulations provide an initial decision-support baseline that can be progressively improved over time, rather than a fixed or static forecast."

---

## [Editor Report · Decision Letter 2]

9 Feb 2026

Investment Modeling for Scalable Agricultural Learning

PONE-D-25-41099R2

Dear Dr. Pittendrigh,

We’re pleased to inform you that your manuscript has been judged scientifically suitable for publication and will be formally accepted for publication once it meets all outstanding technical requirements.

Kind regards,

Yang (Jack) Lu, PhD

Academic Editor

PLOS One
---

## [Editor Report · Acceptance letter]

PONE-D-25-41099R2

PLOS One

Dear Dr. Pittendrigh,

I'm pleased to inform you that your manuscript has been deemed suitable for publication in PLOS One. Congratulations! Your manuscript is now being handed over to our production team.

Kind regards,

on behalf of

Dr. Yang (Jack) Lu

Academic Editor

PLOS One